# Suppressive Antibiotic Treatment in Prosthetic Joint Infections: A Perspective

**DOI:** 10.3390/antibiotics10060743

**Published:** 2021-06-19

**Authors:** Javier Cobo, Rosa Escudero-Sanchez

**Affiliations:** Infectious Disease Department, Hospital Ramón y Cajal, IRYCIS, Ctra. Colmenar Viejo, 28034 Madrid, Spain; javier.cobo@salud.madrid.org

**Keywords:** suppressive antibiotic treatment, prosthetic joint infection, prolonged antibiotic

## Abstract

The treatment of prosthetic joint infections (PJIs) is a complex matter in which surgical, microbiological and pharmacological aspects must be integrated and, above all, placed in the context of each patient to make the best decision. Sometimes it is not possible to offer curative treatment of the infection, and in other cases, the probability that the surgery performed will be successful is considered very low. Therefore, indefinite administration of antibiotics with the intention of “suppressing” the course of the infection becomes useful. For decades, we had little information about suppressive antibiotic treatment (SAT). However, due to the longer life expectancy and increase in orthopaedic surgeries, an increasing number of patients with infected joint prostheses experience complex situations in which SAT should be considered as an alternative. In the last 5 years, several studies attempting to answer the many questions that arise on this issue have been published. The aim of this publication is to review the latest published evidence on SAT.

## 1. Therapeutic Options for Prosthetic Joint Infections

The goal of treating a prosthetic joint infection (PJI) is to eradicate the infection and to maintain or regain implant function. This often involves the replacement of the prostheses, although in some cases (acute infections), the original implant can be salvaged through extensive debridement and prolonged antibiotic therapy, which is referred to as DAIR (debridement, antibiotics and implant retention) [1]. In the remaining situations, the cure can be obtained only by removing the implant, followed by the placement of a new prosthesis, either during the same surgical procedure (one-stage revision) or after a period with antibiotics (two-stage revision) [2]. However, reimplantation is sometimes not possible after removal (resection arthroplasty), and in rare situations, amputation may be necessary. Eventually, due to the patient’s conditions or the anticipated sequelae of the intervention, a potentially curative surgical intervention is waived. In this scenario, orthopaedic surgeons turn their gaze to infectious disease (ID) consultants. Can antibiotic treatment help the patient?

## 2. Concept and Definition of Suppressive Antibiotic Treatment (SAT)

The term "suppressive antibiotic treatment" (SAT) refers to the administration of antibiotics in the long term or indefinitely over time. In the area of PJI, SAT is considered a “noncurative” strategy, in which antimicrobials are administered with the aim of reducing symptoms and delaying or preventing the progression of PJI that needs a surgical procedure to be cured that, for some reason, will not be performed (at least for a prolonged period of time). SAT can also be used in situations in which adequate surgical treatment is performed and the probability of cure is considered very low.

## 3. SAT Indications

SAT appears to be an infrequent therapeutic option in a series (5–14%) that reports the approach of patients with PJI [3,4,5]. However, in those patients over 80 years of age, the percentage treated by SAT can reach 36.5% [6].

SAT is intended to reduce local symptoms (presence of a sinus tract, inflammation, pain, etc.) and thus delay or elude a surgical intervention that has been rejected or is intended to be avoided. It is possible that SAT may delay or prevent prosthetic loosening by reducing the local peri-implant inflammatory process, although no studies have evaluated this potential effect. Additionally, SAT can be considered a general benefit for the patient’s health as a result of the reduction in persistent chronic inflammation [7].

In summary, SAT can be considered for patients with acute PJI for whom conservative treatment (DAIR) has failed, or for patients with chronic-late PJI whose implants are not going to be removed or replaced due to any of the following circumstances:Unacceptable anticipated functional results.Surgical sequelae (or risks) disproportionate to the symptoms.Presence of another disease or condition that makes it advisable to substantially delay the intervention.Short life expectancy.Major surgical contraindication.Patient’s refusal of the intervention.

These situations would therefore be considered PJI with “certain” treatment failure. This would mean that there is evidence of PJI with no curative treatment planned.

There are other situations in which the probability of failure of surgical-medical treatment can be anticipated to be high, although not certain [8,9]. Here, we would cite the following scenarios:Chronic PJI managed with partial replacement of components.Early PJI managed with DAIR and high risk of failure (or potential serious consequences thereof), such as immunosuppressed patients on chemotherapy, patients managed by arthroscopic debridement and/or without replacement of modular components, and cases with suboptimal antimicrobial therapy (multidrug-resistant organisms).Multiple previous failures of treatment of PJI

Once the indications are established, certain conditions are required to be able to carry out SAT:Known aetiology (not essential but lack of knowledge clearly hinders decision-making).Possibility of monitoring and clinical control of adherence and toxicity.Availability of orally active antibiotics against the causal aetiological agent (although, as we will see later, there may be alternatives).

## 4. Evidence on SAT Efficacy

### 4.1. Does SAT Truly Work? What Results Does It Offer?

Evidence of the efficacy of SAT is scarce. A cohort study in which patients with stable PJI (69% with implants for <90 days) were managed with implant retention and prolonged antibiotic therapy for more than 1 year showed that the failure rate (recurrence of infection or need for surgical revision) was four times higher in patients who discontinued antibiotic treatment [10]. Interestingly, most of the patients with discontinued treatment did not exhibit treatment failure, suggesting that many were actually cured. However, the higher rate of treatment failure in patients who stopped taking antibiotics indicates that, in this series, a proportion of patients not cured by DAIR benefited from continuing antibiotic treatment, via delayed or avoidance of failure, which occurred mostly in the first four months. Further arguments in favour of SAT efficacy are provided by the cases that were “rescued” through SAT after the failure of other strategies [10,11,12], as well as by the observation that some SAT failures were temporarily related to the suspension of antibiotic treatment [13].

The interpretation of SAT efficacy is very difficult for three reasons: the absence of controlled studies, the inclusion of patients with acute infections who would be cured by DAIR, and differences in the criteria for evaluating efficacy in published series (Table 1). For example, for some authors, the efficacy criterion was to avoid surgery (even if infection was not controlled) [3], while others required, in addition, control of the symptoms [4,9,11,14]. Success rates varied in the different series from 23% to 84%. However, the series with the highest success rates included patients with early PJI [4,9,14], many of whom would have had the same outcome with much shorter treatments.

We found only one controlled study where patients with PJI at high risk of failure after surgery (DAIR or replacement) managed with SAT were compared with patients in the same conditions who were not managed with SAT. The cases were "matched" using a propensity score. Patients who received SAT had a better outcome at 5 years (68.5% free of infection) than those who did not receive SAT (41.1%) [16]. In a recent multicentre cohort that represents the largest series published to date, we estimated that SAT was effective (control of symptoms and no reintervention) in approximately 75% of the patients after two years and in 50% of patients at 5 years of follow-up [19]. Only patients with persistent infection from whom the implant was not removed were included in this cohort.

### 4.2. What Factors Are Associated with SAT Failure?

Few studies have analysed the factors associated with SAT failure. The failure rate seems higher among patients with a sinus tract and in those with infections caused by S. aureus [13,20,21,22].

In the multicentre study mentioned above, we investigated predictors of failure (defined as the persistence of uncontrolled symptoms of PJI, including sinus tract, or the need for further surgery for debridement or removal of the prosthesis due to infection) [19]. A multivariate analysis showed that the factors associated with failure were the following:Aetiology of infection other than Gram-positive cocci (essentially Gram-negative rods, fungi, or negative cultures). This could be explained because, in general, we have fewer orally active antimicrobials for Gram-negative bacilli.Location of the prosthesis in the upper limbs. It is difficult to explain this finding. In any case, the number of PJIs in the upper limbs was very low.Age less than 70 years. It seems paradoxical, but perhaps younger patients managed by SAT could be more often immunosuppressed or have “tumoural” prostheses, which has been associated with the worst prognosis [17].

In our opinion, at this moment, there are no firm or clear predictors of failure, which means that SAT should not be excluded if the patient meets the conditions mentioned above.

### 4.3. Why Could SAT Stop Working? Is the Development of Resistance Frequent?

In our previously cited cohort study, the coinvestigators were unable to attribute the failure to any specific cause in 52% of the cases. Among the known or attributable causes, the most frequent was the abandonment of treatment or poor adherence (24% of all failures). The development of resistance was not a common cause, as it could only be invoked as a cause of failure in 12% of the cases. This observation has also been made by other authors [18]. In another 11% of patients, the cause of failure was the existence of a previously unsuspected pathogen in cultures that was not covered by the prescribed SAT [19].

## 5. Practical Aspects of SAT

### 5.1. Is a Debridement Mandatory before Starting SAT?

It seems reasonable to think that the reduction in the inoculum and the debridement of infected tissues favours the success of SAT. In most of the series, patients undergo debridement surgery before starting SAT [21]. The difficulty arises in stable patients who present few symptoms, especially if the surgical risks are high. Thus, in the series of SAT in elderly patients, only 24% were operated on [10].

In our analysis, the failure of the SAT was not associated with the absence of a previous debridement [19]. However, surgical debridement makes it possible to obtain valuable samples for microbiological culture, which is a relevant advantage since culture from sinus tracts is not usually representative of the actual aetiology [23].

### 5.2. What Are the Most Suitable Antibiotics for SAT? Is a Combination of Antibiotics Necessary?

From the analysis of the data available in the literature, it is not possible to infer recommendations. The most widely used antibiotic regimens in published series have been the combination of tetracyclines and rifampicin (the last cannot be used alone because of development of resistance) or monotherapy with a beta-lactam or tetracycline antibiotic [3,4,10,14]. In a recent survey of orthopaedists and ID consultants who prescribed SAT, 74% stated that they did not use rifampicin [24].

Since SAT is intended to reduce symptoms and local inflammation, which can be achieved by reducing the bacterial load, antibiotics with activity against stationary growing bacteria are probably not indispensable. In fact, monotherapy with beta-lactams was associated with better outcomes in a large series [10]. It seems reasonable to prioritize tolerability and therapeutic compliance, and for this, it is easier to use monotherapy. In the vast majority of cases, SAT is carried out with orally administered antibiotics. However, there are some recent experiences with intravenous dalbavancin, which have taken advantage of the fact that this drug can be administered once per week or even every two weeks [25], and with the use of beta-lactams such as ceftriaxone or ertapenem subcutaneously [26].

There are no studies on the optimal dosage of antibiotics in SAT. In general, low doses should not be used initially, at least until a reduction in inoculum has been achieved. However, the risks of each antibiotic–bacteria pair must be taken into account. For example, a low dosage of quinolones poses a risk of resistance selection in both staphylococci and Gram-negative bacilli; however, beta-lactam susceptible staphylococci should not develop resistance to a low dose of oral cephalosporins.

### 5.3. Is Intravenous Treatment Necessary at the Beginning of SAT?

Similarly, published studies do not provide an answer to this question. In almost all published series, patients receive several weeks of initial intravenous treatment, but in the aforementioned survey, most of the respondents stated that they do so only occasionally [24].

### 5.4. Can There Be Periods Without Treatment?

The series in the literature reviewed do not include antibiotic treatment-free periods in their protocols. In fact, in some series, failures are reported coinciding with the interruption of treatment, which, in general, appears in the first 4 months after suspension [9].

## 6. Safety of SAT

Information on the safety of prolonged antibiotic treatments can be obtained, not only from studies on SAT in PJI or other osteoarticular infections but also from other areas, such as antibiotic prophylaxis in immunosuppressed patients, the management of specific infections that require very long treatments (multidrug-resistant tuberculosis, actinomycosis, mycobacteriosis, Coxiella endocarditis, etc.) or entities in which infection and bacterial colonization play a relevant role in the natural history of the disease (cystic fibrosis, acne, suppurative hidradenitis, etc.), for which long-term treatments have been tried.

In SAT series, adverse effects are not uncommon, but they rarely require discontinuation of treatment [19,21,22]. In addition, in many cases, poorly tolerated antibiotics can be substituted for another [10,17]. Data collection on adverse effects has not been systematized in any of the published studies and it was always retrospective. Gastrointestinal disturbances and skin reactions appear to be the most common reported adverse events. It should be borne in mind that in most series, ID consultants with extensive experience in the management of antimicrobials are those who prescribe and monitor treatments. Surprisingly, *C. difficile* infection is an infrequent event despite very long treatments that last many years [19,21].

In a preliminary study including several patients on SAT, colonization by multidrug-resistant bacteria was not common. However, the patients who developed infections did so due to bacterial resistance to the antibiotic that they received for SAT [27].

## 7. Reflections and Conclusions

The information on SAT is fragmentary, heterogeneous and of low evidence. Despite this, the analysis of the available series suggests that SAT may represent an option with acceptable efficacy for selected cases in which potentially curative surgery cannot be performed or where the probabilities of success of the treatment are low. It is possible to administer antibiotics safely in the long term, provided that the clinician has the appropriate knowledge and experience. More studies are needed to answer the many questions that remain unanswered. To form useful conclusions in future investigations, it would be desirable to establish pragmatic criteria for efficacy, as well as to separate the cases in which SAT is indicated as an alternative to surgical treatment from those where it is indicated due to a high risk of failure of the surgical treatment used.

## Figures and Tables

**Table 1 antibiotics-10-00743-t001:** Published Series on SAT in PJI.

Reference	Number of Patients	Type of Infection	Aetiology (%)	Follow-Up (Months)	Criteria for Success	Success Rate	Toxicity
Goulet, 1988 [3]	19	90% chronic 10% acute	*S. aureus* (21%), CoNS (21%), *Streptococcus* spp. (32%)	49.2	Retention of the implant	63%	No data
Tsukayama, 1991 [15]	13	100% chronic	*S. aureus*, (54%), CoNS (46%)	37.2	Retention of the implant	23%	38% antibiotic needed to be changed
Segreti, 1998 [4]	18	50% chronic 50% acute	*S. aureus* (44%), CoNS (44%)	48	Remained asymptomatic and functional prosthesis	83%	22% CDI
Rao, 2003 [14]	36	53% chronic 47% acute	*S. aureus* (26%), CoNS (50%)	60	Remained asymptomatic and functional prosthesis	86%	8% diarrhoea
Marculescu, 2006 [13]	88	No data	*S. aureus* (32%), CoNS (23%)	23.3	Absence of the following: Relapse, reinfection, presence of acute inflammation in the periprosthetic tissue or at any subsequent surgery on the joint, development of a sinus tract, death from prosthesis-related infection, or indeterminate clinical failure	57%	3% diarrhoea, 11% hypersensitivity, one case of CDI
Byren, 2009 [9]	112	31% chronic 69% acute	*S. aureus* (40%)*,* CoNS (23%)	27.6	Absence of the following: Recurrence, wound or sinus drainage recurring or persisting for 3 months beyond the index debridement procedure or requirement for revision surgery (irrespective of the indication)	82%	No data
Prendki, 2014 [6]	38	61% chronic 39% acute	*S. aureus* (39%), *Streptococcus* spp. (18%), Gram-negative bacilli (17%)	24	Absence of the following: Persisting infection, relapse, new infection, treatment discontinuation because of severe adverse events, or related or unrelated death	60%	1 case of recurrent CDI.
Siqueira, 2015 [16]	92	61% chronic 39% acute	*S. aureus* (48%), CoNS (35%)	69.1	Absence of the following: Subsequent surgical intervention for infection after the index procedure, persistent sinus tract, drainage, or joint pain at the last follow-up visit, or death related to the PJI	69%	No data
Prendki, 2017 [10]	136	No data	*S. aureus* (62%), CoNS (21%)	24	Absence of the following: Local or systemic progression of the infection, death, or discontinuation because an adverse drug reaction	61%	18.4% discontinued antibiotics, but in half of cases, the antibiotic could be replaced by another.
Pradier, 2017 [8]	39	61% delayed or late	*S. aureus* (79%), CoNS (10%)	24	Absence of the following:	74%	15% (phototoxicity and gastrointestinal intolerance)
		39% acute			Signs of infection assessed ≥24 months after the end of the curative treatment and then at the last contact with the patient, or death related to the PJI		
Wouthuyzen- Bakker, 2017 [17]	21	62% late or delayed 38% early	*S. aureus* (33%), CoNS (38%)	21	Absence of the following: Pain during follow-up, surgical intervention is needed to control the infection, or death related to PJI	67%	43% reported side effects and needed change or adjustment of the dosage.
Pradier, 2018 [18]	78	60% delayed or late 40% early	*S. aureus* (40%), CoNS (32%)	34	Absence of the following: Signs of infection assessed ≥24 months after the end of the curative treatment and then at the last contact with the patient, or death related to the PJI	72%	18% phototoxicity and gastrointestinal disturbance
Escudero-Sánchez, 2019 [19]	302	73% chronic 11% haematogenous 16% early postoperative	*S. aureus* (31%), CoNS (33%)	36.5	Absence of the following: Appearance or persistence of a sinus tract, need for debridement or replacement of the prosthesis due to persistence of the infection, or the presence of uncontrolled symptoms, death related to PJI	59%	17% gastrointestinal 5% cutaneous
Leijtens, 2019 [20]	23	30% early 70% late or delayed	*S. aureus* (2%), CoNS (61%)	33	Absence of the following: Reoperation for PJI or death related to PJI	56.5	24% needed change or dosage modifications.
Sandiford, 2019 [5]	24	No data	*S. aureus* (25%), CoNS (21%)	38.4	Absence of the following: Sepsis arising from the affected joint, no progression to further surgery, or death related to PJI.	83	4.2% rash 4.2% rifampicin interaction

CDI: Clostridioides difficile infection; CoNS: coagulase-negative staphylococci.

## Data Availability

The data presented in this study are available on request from the corresponding author. The data are not publicly available due to the fact that it is a multicenter study and the complexity of the clinical history in some centers.

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
