# Peer review of "Suppressive Antibiotic Treatment in Prosthetic Joint Infections: A Perspective"

_antibiotics, 2021, doi:10.3390/antibiotics10060743_

Round 1

Reviewer 1 Report

The manuscript is a review of the use of suppressive treatments in PJI. The review is well written and includes some interesting issues that could be helpful for the specialized reader. Some minor comments:

1-5.2: It could be useful to add a short phrase against the use of rifampin in monotherapy (line 174) because it could select resistant clones.

2-6: A paragraph with a detailed description of the common side effects could be helpful.

Author Response

1-5.2: It could be useful to add a short phrase against the use of rifampin in monotherapy (line 174) because it could select resistant clones.

  • It has been added (in brackets):

The most widely used antibiotic regimens in published series have been the combination of tetracyclines and rifampicin (the last cannot be used alone because of development of resistance) or monotherapy with a beta-lactam or tetracycline antibiotic.

2-6: A paragraph with a detailed description of the common side effects could be helpful.

  • The next sentence has been added:

Data collection on adverse effects has not been systematized in any of the published studies and it was always retrospective. Gastrointestinal disturbances and skin reactions appear to be the most common reported adverse events

Reviewer 2 Report

This is a comprehensive review of an approach to a difficult clinical situation. It encompasses all published literature, and analyses it well.

The authors should consult someone with English as the first language, as the paper needs styling and proper use of terms.

Author Response

We apologize for any errors that may be in the article. It was reviewed by a native English speaking editors. We can provide the edition certificate, if necessary.

Reviewer 3 Report

In the review article entitled “Suppressive antibiotic treatment in prosthetic joint infections:  a perspective” Cobo et al describes in detail about suppressive antibiotic treatment in prosthetic joint infections. The article is well structured and authors managed to gather detailed information on a relevant but rarely discussed topic in orthopedics. However, I have few comments or suggestions for the authors to consider:

  1. Abstract: Authors could consider adding a sentence about the aim/focus of this review.
  2. Line 32-33: Is it only antibiotic treatment or long term/indefinite antibiotic treatment?
  3. Line 35-42: Are there any specific definition on SAT by MSIS, IDSA or ICM on PJI?
  4. Line 36-37: Is there any minimum time period for SAT?
  5. Line 44-82: It would be nice to have a table or figure with SAT indications.
  6. Line 120-121: All bacterial names should be in italics.
  7. Line 168-184: Describe in brief about the antibiotics for SAT recommended by IDSA, ICM on PJI or other orthopedic infection groups.

Author Response

  1. Abstract: Authors could consider adding a sentence about the aim/focus of this review.
  • It has been added a sentence:

The aim of this publication is to review the latest published evidence on SAT.

2. Line 32-33: Is it only antibiotic treatment or long term/indefinite antibiotic treatment?

  • The question is posed from the orthopedic surgeon's perspective as a generic question. In the context in which it is formulated implicitly, we refer to the suppressive treatment, as seen in the following paragraph.

3. Line 35-42: Are there any specific definition on SAT by MSIS, IDSA or ICM on PJI?

  • Suppressive treatment is mentioned in the IDSA guidelines as a possible strategy even after appropriate surgeries. No definition is provided in the ICM documents, although there is high consensus in the statement that it can be used when it is considered that the PJI surgery has not been appropriate to obtain a PJI cure, and it is said that the duration should be lifelong, but always individualizing the cases.

  1. Line 36-37: Is there any minimum time period for SAT?
  • In our opinion a specific duration (for example 3 months or 6 months) should not be considered as "suppressive" but as prolonged therapy. The term suppressive implies that cure is not possible with "defined" treatment. This does not mean that a suppressive treatment is used in a non-indefinite way in a specific patient for any reason.

  1. It would be nice to have a table or figure with SAT indications.
  • We think that being highlighted by points is perceived well. In any case, we submit to what the editor decides about constructing a table with the two types of indications

6. Line 120-121: All bacterial names should be in italics.

  • Thank you, it has been changed.

7. Line 168-184: Describe in brief about the antibiotics for SAT recommended by IDSA, ICM on PJI or other orthopedic infection groups.

  • We thank the suggestion. It seems appropriate to add a paragraph on the dosage of antibiotics for SAT, but obviously there are no studies that can allow to establish recommendations. A table with some dosages of oral antibiotics in the normal or low dose ranges is provided in the IDSA guidelines. The next paragraph has been added

There are no studies on the optimal dosage of antibiotics in SAT. In general, low doses should not be used initially, at least until a reduction in inoculum has been achieved. However, the risks of each antibiotic-bacteria pair must be taken into account. For example, a low dosage of quinolones poses a risk of resistance selection in both staphylococci and gram-negative bacilli; however, beta-lactam susceptible staphylococci should not develop resistance to a low dose of oral cephalosporins.